# Changes in Quantity Measures of Various Forms of Cannabis Consumption among Emerging Adults in Canada in Relation to Policy and Public Health Developments

**DOI:** 10.3390/ijerph20136213

**Published:** 2023-06-24

**Authors:** Susan J. Yousufzai, Adam G. Cole, Mika Nonoyama, Caroline Barakat

**Affiliations:** Faculty of Health Sciences, Ontario Tech University, Oshawa, ON L1H 7K4, Canada; adam.cole@ontariotechu.ca (A.G.C.); mika.nonoyama@ontariotechu.ca (M.N.); caroline.barakat@ontariotechu.ca (C.B.)

**Keywords:** cannabis, COVID-19, emerging adults, EVALI, forms, quantity, legalization

## Abstract

Limited research examines changes in quantities of various forms of smoked/vaped cannabis among regular consumers, including emerging adults (EAs; 18 to 29) in Canada. This information is particularly relevant in the current context of emerging cannabis behaviors among EAs related to political amendments (legalization of cannabis), vaping-related lung illnesses (EVALI), and unprecedented pandemics (COVID-19). This study investigated the impact of legalizing recreational cannabis use in Canada, the EVALI epidemic, and the COVID-19 pandemic on the quantity of smoked/vaped forms of cannabis in relation to gender differences. EAs retrospectively self-reported the quantity of herb, hash, concentrates, joint size, and the number of joints and vaping cartridges in relation to three consecutive developments: pre-legalization, post-legalization; pre-EVALI, post-EVALI, pre-COVID-19, and during COVID-19. The quantity of herb use significantly increased among heavy users, and vaping quantity significantly increased among light users. Overall, an increasing incremental trend was observed in the average quantity of cannabis forms used over time. Males consumed higher quantities of all cannabis forms than females. More males than females reported using concentrates (*p* < 0.05). These findings reveal unique aspects of the amount of various cannabis forms smoked/vaped in relation to gender and provides preliminary evidence of cannabis consumption behaviors in relation to changing social and cultural contexts.

## 1. Introduction

In Canada, cannabis consumption rates are characterized by marked differences in relation to age and sex [1,2]. For example, the prevalence of daily or almost daily consumption is higher (12.5%) among emerging adults (EAs; 18 to 24-year-olds) than among 45- to 64-year-olds (4.8%) [3]. In relation to sex, 7.6% of males compared to 4.5% of females report using cannabis daily or almost daily [3]. In addition, recent evidence shows that almost two-thirds (65%) of all cannabis products in Canada are consumed by a relatively small population (10%) of very heavy cannabis users [4]. This was mostly accounted for by the younger age group (15 to 34), and males in particular [4]. These findings suggest that examining cannabis consumption patterns and related health implications among frequent and heavy consumers (i.e., EAs) in relation to gender may be critical for addressing public health outcomes.

Contrary to its widespread permissive use and its presumptions as a ‘benign’ substance among EAs [5,6], cannabis-related harm is evident when it is used regularly or heavily, and when use is initiated at a young age [3,7,8]. For example, an earlier onset and higher frequency—particularly weekly or daily patterns—of cannabis use puts individuals at an elevated risk for dependency [3,9,10]. However, frequency patterns alone may not reflect the extent of risk [10]. Among users reporting daily use, dependency risk is differentiated by the actual quantity of use (e.g., number of joints smoked per week) [11], suggesting that quantity is a critical measure of cannabis-related risks. However, a clear understanding of cannabis-related behaviors, harms and effects are hindered by the paucity of studies examining various methods of consumption and measuring the quantity of consumption in relation to various cannabis forms [3,4,10,12].

The complexity of measuring cannabis-related effects and risks are further complicated by sex-specific neurobiological mechanisms, gender-specific consumption differences, and motives for use. Nascent evidence suggests a higher sensitivity to cannabis among females, and their vulnerability based on the accelerated progression to problematic use [13], although males’ development of a cannabis use disorder (CUD) is more reliant on gender and social norms that influence earlier initiation, accessibility, and acceptability [14].

Emerging trends of cannabis consumption behaviors and diversified profiles associated with recreational cannabis legalization in Canada since 17 October 2018 warrants research to examine the quantity of cannabis used, as it may vary across products and methods of use [4,15]. The legalization of recreational cannabis in Canada involves controlling the production, distribution, sale, and possession of cannabis for adults 18 years or older after the passing of Bill C-45, having transitioned from the former exclusive medicinal policy since 2001 [3]. Numerous other countries have also expanded their legal access to cannabis under this regime (e.g., Uruguay and some U.S. states) and other different regulatory frameworks, by decriminalizing possession or enacting tolerance policies [16]. Decriminalisation is the reduction of penalties for cannabis use while maintaining penalties for cannabis supply [16]. With respect to tolerance policies, in the Netherlands, while it is against the law to possess, sell or produce drugs, coffee shops are permitted to sell cannabis under certain strict conditions [17]. Nations like Singapore, where cannabis continues to be prohibited, have stringent regulations in place to combat the trafficking, possession, use, and importation or exportation of illegal drugs, which encompasses cannabis and its derivatives [18].

With increasing tolerance, and liberalization, there have been essential changes and global trends in the pattern and type of cannabis use which have occurred, while the prevalence of cannabis use has increased overall in the past 15 years. During the year 2020, over 209 million individuals, which accounts for more than 4 percent of the global population aged 15 to 64, reported using cannabis [19]. Comparatively, the prevalence of past-year cannabis use has risen by 8 percent from 2010, when it stood at 3.8 percent [19]. In reflection of cannabis-related liberalization policies in other parts of the world, Canadian researchers predicted that the legalization of recreational cannabis in Canada would have a similar overarching influence on consumption [20]. Research conducted in the United States, focusing on jurisdictions like Oregon and Colorado, has indicated that the legalization of recreational cannabis use has contributed to higher rates of cannabis consumption among undergraduate students [21,22]. Specifically, after recreational legalization in Colorado, Parnes et al. [22] observed an increase of nearly 11% in the rate of cannabis experimentation (from 43.5% to 53.6%) (*p* < 0.001). This increase was more prominent among students aged 21 and above (the legal age for possession), rising from 40.4% to 60.9% (*p* < 0.001). Similarly, according to recent nationwide data from Canada, there was an increase in cannabis use between 2018 and 2019, specifically among individuals aged 25 and older (from 13.1% to 15.5%) and males (from 17.5% to 20.3%) [23]. However, the rates for 15- to 24-year-olds (from 27.6% to 26.4%) and females (from 12.3% to 13.4%) remained steady during that period [23]. Consequently, the trend towards increased cannabis consumption is expected to continue as accessibility improves, social acceptance grows, and perceptions of harm diminish [22,24].

Simultaneously, the successional and unprecedented global developments of specific public health events may have influenced trends in cannabis use and patterns of exposure. Following the federal legalization of recreational cannabis use in Canada, subsequent public health developments have raised concerns in relation to the health outcomes associated with behavioral changes in cannabis consumption among at-risk and high-consumer groups [25,26]. This includes the ‘electronic cigarette or vaping product associated lung injury’ (EVALI) outbreak, and the spread of severe acute respiratory syndrome coronavirus 2 (SARS-CoV2) disease or ‘COVID-19’. Indeed, findings from one study found that such developments have influenced changes in the frequency of smoking or vaping cannabis among EAs [27]. In light of the respiratory risks associated with EVALI and COVID-19, using methods of combustion and vaping to consume cannabis have been considered risk factors that may exacerbate complications associated with a COVID-19 infection and increase susceptibility to lung injury [28,29]. 

The drug reform of legalizing cannabis for recreational use has been implemented with a cannabis policy framework that aims to maintain strict health-focused regulation. Despite advocated benefits, such as reducing illicit markets, and protecting youths by reducing access, challenges remain in attempting to control factors such as the increasing prevalence of problematic cannabis use [24,25]. Specifically, the access to higher-potency cannabis forms, new modes of administration, and reduced risk perception of individuals previously practicing caution due to its former illicit status may gauge curiosity [30]. For example, evidence suggests a growing trend of using cannabis concentrates as a result of legalization across the U.S. [31]. These forms typically contain 60–85% ∆-9-tetrahydrocannabinol (THC) content, compared to 10–12% found in plant material [31]. Such forms have been linked to mental health problems, and dependence [32,33]. On the other hand, legalization has coincided with ample opportunity to produce novel modes such as electronic vaping devices, which can increase appeal, and pose reasonable concerns due to limited understanding of respiratory risks associated with the wide variety of vaping products and compounds [34,35].

Different inhalation methods of cannabis such as joints, blunts, pipes, bongs, dab rigs, and vaporizers, are used for different forms of cannabis and can vary with the quantity used per dose and the potency of THC [15,36]. While smoking traditional forms of cannabis such as dried herb remains the most common method of consumption, concentrates (e.g., “shatter”, “budder”), oils and hash, are becoming increasingly popular, which also contain higher THC concentrations [3,4]. Simultaneously, novel iterations of vaping devices in addition to smoking (dual use) are being used to vape these diverse arrays of cannabis forms [37,38]. Indeed, vaping ‘carts’ is viewed as a convenient and appealing method of consumption, especially among EAs [39]. However, vaping may pose significant health risks for cannabis users. In fact, when the Center for Disease Control and Prevention (CDC) termed the epidemic of EVALI in late 2019 (approximately November) [40], 82% of cases in the U.S. were found to be caused by a compound mixture of THC vaping oil and vitamin-E-acetate. A proportion of 78% of these cases reported informal sources of vaping products (such as friends, online or dealers) [41].

In Canada, between September 2019 and December 2020, 20 cases of vaping-associated lung illness were reported [42]. The incidences of lung injury associated with vaping in the US have surmounted the numbers in Canada, with more than 2800 cases and 68 deaths [35,43]. Overall, young adults are the most affected population [35,43]. A common symptom reported by the 20 patients in Canada included respiratory symptoms, and 75% (15 of 20 cases) reported a combination of respiratory, gastrointestinal (e.g., nausea, diarrhea), constitutional (e.g., chills, fatigue) and/or other symptoms (e.g., fever, poor appetite/weight loss) [42]. These prognoses also made it difficult to differentiate the heterogenous pathology of EVALI symptoms from COVID-19, during a period in time in which both infections were simultaneously prevalent [44]. It is important to note that cannabis containing vapes were legally available 1 year later (October 2019) in Canada, which also aligns with EVALI [42]. In addition, a parallel increase was found in relation to past 30-day use of vaping devices in 2019 [42]. Among EA, the use of ever-vaping cannabis-containing products (7%) exceeded that of youths and adults (3% each, respectively) [42]. Since THC-based products have been speculated to be linked with the EVALI outbreak (due to the specific use of additives such as vitamin E acetate), it is important to examine these changes among cannabis vapers and how it may influence consumption patterns based on the perceived safety of combusted compared to vaporized cannabis products.

Some evidence suggests that there have been declines in e-cigarette sales following news of the multistate EVALI outbreak in the U.S. (September 2019–January 2020) which may have influenced cessation intentions of e-cigarette and vaping-product use [45], and increased risk perceptions among adolescents [46]. Among adult users in the U.S., one study found that the EVALI outbreak had limited impacts on decreasing the frequency of THC vaping [47]. However, there are no studies to our knowledge examining subsequent changes in the quantity of consumption in relation to the vaping of cannabis among EAs. These findings warrant research that examines how EVALI may influence cannabis-specific vaping and smoking consumption, even in countries such as Canada that have fewer EVALI cases [17]. In addition, the lack of understanding about the health effects of various constituents when aspirated provides significant reason for the continuous monitoring of consumption behaviors [34,35].

The bi-directional relationship between COVID-19 and substance misuse has been a key concern since the declaration of the pandemic on 11 March 2020 [48], especially due to the increased vulnerability to adverse respiratory consequences associated with smoking or vaping [28,49]. Evidence suggests increases in cannabis frequency (days used) among adolescent girls in particular [50], and more EAs smoking a greater number of joints on a daily basis in other regions [51]. Studies have not reported the impact of the COVID-19 pandemic on the quantity of various forms of cannabis used (e.g., herb, hash, concentrates, oil), although recent data collected in 2021 from the Canadian Cannabis Survey revealed an increase in the amount of cannabis usage due to COVID-19. More people (46%) aged 16 to 19 years and 20 to 24 (40%) specifically reported an increase, compared to those 25 years or older (25%) [52]. Another study by Sznitman et al. [53] found that secondary stressors were associated, both directly and indirectly, with increased cannabis use during COVID-19 among adults in Israel.

Changes in consumption may be unique for EAs in Canada, who have legal access to a diverse range of cannabis products. In addition, during the pandemic, EAs may have been more prone to use cannabis to allay negative feelings, often noted as ‘coping-oriented reasons’ that increase the risk of substance dependence [9]. Moreover, given that EAs use cannabis as a social cohesion tool, and sharing joints and vaping devices is common practice among cannabis users [51], social restrictions may have affected individual quantity of consumption [54].

The aim of this study was to examine if there was a change in the retrospective self-reported quantity of cannabis consumption forms (dried herb, concentrates, hash, and vaping oil/juice) following the legalization of recreational cannabis use in Canada, the EVALI epidemic, and COVID-19 pandemic, among male and female EAs in Canada.

## 2. Materials and Methods

### 2.1. Study Design and Ethical Approval

This is a cross-sectional exploratory study. Ethics approval for this research was received from the Ontario Tech University Research Ethics Board on 4 May 2020 [REB#15880].

### 2.2. Participants and Recruitment

Participation was limited to the following inclusion criteria: reported smoking and/or vaping cannabis in the last 12 months, currently living in Canada at the time of the study, and 18 to 29 years old. The survey was restricted to the English language, and consent was required to participate. Respondents that met the inclusion criteria and consented were then directed to the survey questions, which was designed to take 20 to 25 min to complete. No compensation was provided. All responses were anonymous. Convenience sampling was used to recruit participants 18 to 29 years old through social media platforms (Facebook and Twitter) across Canada and from Ontario Tech University. Recruitment and data collection occurred from 4 May 2020 to 31 August 2020. Respondents were identified as cannabis users if they reported smoking and/or vaping cannabis in the last 12 months. Results of 343 individuals were reviewed, of which 312 were eligible. Respondents were excluded if they were not 18–29 years old (*n* = 11), did not smoke or vape cannabis in the last 12 months (*n* = 13) (e.g., only used edible forms), or did not consent (*n* = 7).

### 2.3. Measures

#### 2.3.1. Independent Variables

To examine any changes in the quantity of cannabis forms reported in this study, participants were asked to retrospectively report how each of the following six time periods that were marked by major legislative and public health developments influenced their cannabis use: (1) Prior to Legalization (approximately October 2018); (2) since legalization; (3) prior to EVALI (approximately November 2019); (4) since EVALI; (5) prior to imposed COVID-19 preventative measures (approximately March 2020); (6) since imposed COVID-19 preventative measures.

#### 2.3.2. Dependent Variables

Questions pertaining to forms and quantity measures of cannabis used were derived and modified from the *International Cannabis Policy Study* (ICPS) [55], and *Daily Sessions, Frequency, Age of Onset, and Quantity of Cannabis Use Inventory* (DFAQ-CU) [56]. The assessment of cannabis quantity forms included: (1) dried herb; (2) hash; (2) concentrates; and (3) cannabis vaping oil/juice. For each form of cannabis, participants were asked to report if they considered it as one primary form that they used predominantly or most often (DFAQ-CU): “Do you consider dried herb as one ‘primary’ form of cannabis that you use?”; “Do you consider Hash/Hashish as one ‘primary’ form of cannabis that you use?”; “Do you consider cannabis concentrates (e.g., Wax, Shatter, Butane Hash Oil, Dabs) as one ‘primary’ form of cannabis that you use?”; “Do you consider cannabis oil/vape juice as one ‘primary’ form of cannabis that you use?”. Participants were given response options: “yes” or “no”. Answering “yes” directed participants to the specific form used and corresponding quantity measures. Participants may have chosen more than one form to report. For each primary form selected, participants reported retrospectively the quantity smoked/vaped prior to and since each event.

The quantity of cannabis in dried herb form consumed before and after each event were based on the following questions: “On the days you use(d) the dried herb, about how much did you personally use?”; “On the days you use(d) joints, what joint size is closest to the size you normally smoke(d)?”; “On the days you use(d) joint(s), how many did you smoke?”. For herb quantity, participants were able to choose the method to quantify their use, by reporting the amount of dried herb used in grams or the equivalent joint size in grams and number of joints personally smoked. Participants were queried: “Is it easier for you to tell us the joint size or the amount of dried herb (e.g., grams or ounces) you use?”. Images of herb quantities and joint sizes with corresponding measurements were provided for reference in the survey to help quantify the amount used (derived from [55]). Herb quantity was based on the amount measured in grams [0, less than 1/8 g (i.e., 0.0625), 1/8 g, ¼ g, ½ g, ¾ g, 1 g, 2 g, 3 g, 1/8 ounce, ¼ ounce, more than ¼ ounce]. Joint size was measured in grams (0, 0.2, 0.4, 0.6, 0.8, 1.0, 1.2) and the number of joints [i.e., 0, ¼ (0.25), ½ (0.5), ¾ (0.75), 1, 2, 3, 4, 5, 6, 7, 8, 9, 10, and more than 10].

The quantity of hash or concentrate consumed as primary forms were based on the following questions: “On the days you use(d) Hash/Hashish how much did you personally use?”; “On the days you use(d) concentrates how much did you personally use?”. Hash and concentrate quantity were also measured in grams [0, less than 0.5 (0.25), 0.5, 1, 2, 3, 4, 5, 6, 7, more than 7 (e.g., 8)]. Reference images were provided to quantify 1 g of hash and concentrate (derived from [55]).

For vaping cannabis oil/vape juice, participants were asked: “What size vaping cartridge do you ‘primarily’ use?”, and “How many vape cartridge(s) or refill(s) did you use in a month?”. Vaping quantity was determined based on the number of vape cartridges/refills used in a usual month [0, less than half of a cartridge (0.25), ½ (0.5), 1, 2, 3, 4, 5, 6, 7, 8, 9, 10, and more than 10 (e.g., 11)]. Numeric assumptions were inputted for options “less than” or “more than” the amount indicated. Measuring vaping quantity by the number of cartridges over a longer timeframe than a day is found to be effective in measuring consumption. Therefore, the consumption of vaping was examined by the average of vaping cartridges used in a month. Evidence shows most respondents prefer to report the number of cartridges or tank refills (51%), and fewer users finish 1 cartridge/tank in a day [57]. Data for the size of the vaping cartridge used was omitted due to limited responses to the open-ended question.

Participants reported the cumulative lifetime days of smoking or vaping, using the following questions: ‘How many days in your lifetime have you smoked cannabis?’; ‘How many days in your lifetime have you vaped cannabis?’. A binary variable was created to differentiate an occasional user (i.e., light user) from an experienced user (i.e., heavy user) based on lifetime days of use. Occasional users included smoking or vaping: 1 to 2; 3 to 10; or 11 to 99 days in their life. Experienced users included those that reported 100 to 999 or more than 1000 days of lifetime smoking or vaping [38]. Frequency patterns of smoking and/or vaping cannabis on a usual day before and after the occurrence of all six time periods were measured based on categories: (0) less than once a month; (1) monthly; (2) weekly; (3) daily use.

#### 2.3.3. Explanatory Variables

Due to the limited outbreak and awareness of EVALI in Canada, participants were asked, “Prior to taking this survey did you know about EVALI?”. The options were dichotomized (“yes” or “no”). Demographic characteristics included sex (female, male) and postal region. Participants were asked, “what was your sex at birth?”, and provided the first three digits of their postal code (“Please enter the first three digits of your current postal code, e.g., L1V”). Postal codes were grouped by province.

### 2.4. Data Analyses

Data were analyzed using SPSS v 27 (IBM SPSS Statistics 27 ©). Chi-square tests were used to analyze differences in relation to the form of cannabis used most of the time by sex (Figure 1). A one-way repeated measures analysis of variances (ANOVAs) using a Bayesian analysis was conducted to evaluate the effects of the six time periods on the six dependent continuous variables in relation to the quantity of the various forms used: (1) quantity of herb in grams, (2) joint size, (3) number of joints, (4) number of vaping cartridges, (5) quantity of hash in grams, and (6) quantity of concentrates in grams. The Bayes factor was used to indicate statistical significance (*p* < 0.05) using the estimation method for Rouder’s mixed design. The Bayes factor test yields a *BF* quantifying how well the alternative hypothesis (H_1_) predicts the empirical data relative to the null hypothesis (H_0_) [58,59]. The strength of the evidence relies on BFs above 3 for the alternative hypothesis and below 0.33 for the null hypothesis, while values between approximately 0.33 and 3 indicate that the data are insensitive [60]. Analyses stratified by sex for females and males were conducted using a general-linear-model repeated measures ANOVA to describe differences in the quantity of cannabis forms for the six outcome measures. Post hoc comparisons using the Bonferroni correction were conducted to determine significant between-subject differences for each time interval and outcome.

The average quantity of cannabis use per month for each form was calculated by multiplying it by the frequency of smoking/vaping categories to reflect the number of days used per month (0 = less than once a month; 1 = monthly; 4.3 = weekly; 30 = daily) in relation to each time period. Smoking frequency for each time period was multiplied by the herb quantity, joint size and joint quantity reported. Vaping frequency for each time period was multiplied by the number of cartridge refills in order to estimate the average quantity of vaping cartridges used in one month.

The six time periods were divided into four time periods by calculating the average quantity of use between the periods after October 2018, and before November 2019 to create one category (October 2018 to November 2019) given that the quantity of herb, joints and vaping remained consistent in each period. In addition, the average quantity between the two periods—since EVALI (November 2019) and before COVID-19 (March 2020)—were calculated to show the quantity of use from November 2019 to March 2020.

## 3. Results

The final sample included 312 cannabis users (189 female, 123 male). More than three quarters of participants (76%) indicated a postal region from Ontario; few participants were from other Canadian provinces and territories (24%). The most popular form of cannabis used as the primary form was herb for males and females (91%), followed by cannabis vaping oil/vape juice, which was endorsed by 15% of males and 15% of females. More males (15%) than females (7%) reported using cannabis concentrates as the primary form (ꭓ^2^ (1) = 5.02, *p* < 0.05). Hash was indicated as the least primary form used by males (3%) and females (5%) (Figure 1). Only 24% of participants were aware of EVALI before taking the survey, and 62% of these individuals had vaped in the last 12 months.

The majority of the sample comprised highly experienced (exclusive) cannabis smokers, with 63% of males and 54% of females reporting 100 or more lifetime days of smoking. The mean (SD) age of smoking cannabis onset was 17 (±2) years, while the age of vaping onset was higher at 21 (±3) years (only 2% of users had exclusively vaped cannabis in the last 12 months).

### Changes in Quantity of Cannabis Consumption Forms

Table 1 shows the average quantity of cannabis personally used in different forms on a typical day over time for the sample population. There was a statistically significant difference in herb quantity smoked over time in relation to the various events (*BF* (14) = 4.496), *p* < 0.001, showing strong evidence for an effect on the quantity of herb used over time in the sample population (*Bf* > 3) (Table 1). Analyses separated by sex indicated no significant difference for males (*F* (2.35, 176.405) = 0.939, *p* = 0.41), but a statistically significant difference for females (*F* (2.13, 212.941) = 5.585, *p* = 0.004), showing that herb quantity almost doubled from 0.67 g to 1.2 g from pre-legalization to since the COVID-19 pandemic. Post hoc comparisons revealed that a significant increase was evident from before and after legalization for females (*p* = 0.005), in which herb quantity increased from an average of 0.67 g to at least 1 g personally used on a typical day.

Joint quantity and joint size significantly differed over time (*BF* (14) = 65.776, *p* < 0.001) (Table 1). Strong evidence in support of the experimental hypothesis (H_1_) showing an effect on joint quantity over time was indicated (*Bf* > 3). Analyses separated by sex showed that there was a significant increase in joint quantity for males (*F* (2.25, 58.521) = 3.05, *p* = 0.049 and for females (*F* (2.4, 134.6) = 4.902, *p* = 0.006). The average number of joints smoked on a typical day by females was highest during the pandemic period (M = 0.86). In relation to joint size, the overall results indicate no evidence for an effect on joint size in relation to the events (*BF* (14) = 0.903, *p* < 0.001). Although analyses stratified by sex showed no difference for males (*F* (1.81, 45.284) = 1.83, *p* = 0.17), a borderline significant difference was found for females (*F* (3.01,168.578) = 2.67, *p* = 0.049).

In addition, results indicate no difference in concentrate quantity in relation to the events (*BF* (14) = 0.195 *p* < 0.001) (Table 1). Analyses stratified by sex indicated no significant differences for males (*F* (5, 42.54) = 2.84, *p* = 0.40) or females (*F* (1.85, 20.39) = 2.49, *p* = 0.11). Changes in hash quantity also indicate moderate evidence for no effect across the time intervals (*BF* (14) = 0.066, *p* < 0.001). Due to the small subsample of participants using hash, we were unable to stratify based on sex.

An increasing trend in the quantity of average vaping was found over time (*BF* (14, 130.955) = 0.954 *p* < 0.001), although the *Bf* < 3 (Table 1). Analyses stratified by sex indicated a significant increase for females (*F* (2.558, 71.632) = 2.922, *p* = 0.048). Although there was no significant difference for males (*F* (1.57, 25.11) = 1.296, *p* = 0.29), an average increase is seen before and after legalization and since the COVID-19 pandemic, but a decrease from before and after EVALI (Table 1).

Table 2 shows average monthly use for each time period. Herb quantity on average increased significantly over the time periods (*F* (2.47, 425.22) = 9.25, *p* < 0.001) for the total sample and for females specifically (*F* (2.22, 219.69) = 7.86, *p* < 0.001). The quantity of use calculated based on the average joint size and number of joints also significantly increased for the total sample (*F* (1.88, 148.75) = 6.25, *p* = 0.003) and for females (*F* (1.92, 101.78) = 4.06, *p* = 0.022). Table 3 shows the average quantity of cannabis personally used over four time periods for heavy and light users in relation to herb quantity, joint quantity, and vaping quantity. Among heavy users, there was a significant increase in herb-quantity use from before legalization to since COVID-19 (*F* (1.68, 230.02) = 9.03, *p* < 0.001). Joint quantity also increased significantly among heavy users (*F* (1.51, 64.71) = 9.48, *p* < 0.001). There was an increase over time in relation to vaping for both heavy and light users; however, this association was only significant among light users *F* (1.99, 49.83) = 5.89, *p* = 0.005.

## 4. Discussion

This study contributes to the complex and evolving landscape of cannabis consumption by examining changes associated with various quantity measures of cannabis forms consumed through smoking and vaping before and after political amendments (legalization), public health concerns (EVALI) and during unprecedented natural societal health threats (COVID-19). The findings from this study provide insight into possible changes in cannabis consumption among EAs as a result of public health events. The current legislation on recreational cannabis consumption puts EAs in a social environment that allows for easy accessibility and experimentation and makes them vulnerable to health risks. Recent evidence has shown an increase in the prevalence of cannabis users since legalization [23]. The current study contributes to this evidence, suggesting that other measures of consumption, specifically the average quantity used by an individual, may also increase. This was observed among females in particular.

For a clearer estimation of quantity, the quantity measures of the various cannabis forms in this study can be compared to a recent Canadian study that established an algorithm for measuring the quantity of various cannabis forms in standard joint size equivalents [4]. Callaghan et al. [4] developed standard measurements of multiple cannabis forms equivalent to a joint that contains 0.5 g of dried herb—the smallest retail joint size sold in provincial/territorial government cannabis stores. To illustrate, physical production equivalencies across major cannabis products compared to a joint containing 0.5 g of dried herb is equivalent to 0.125 g of hash, 0.096 g of cannabis oil for oil cartridges, and 0.096 g of concentrate [4]. Comparatively, our study shows that on average, users are consuming more than 1 joint-equivalent amount of cannabis per day of dried herb, hash, and concentrate, and within a month of using vaping cartridges. In addition, our study shows that the average joint size is larger than the standard joint size indicated by Callaghan et al. [4]; thus, the quantity of cannabis consumed may be underestimated for the EA population; additional investigation into possible differences in the quantity of cannabis consumed across age groups is warranted.

Overall, the primary findings of this study suggest that cannabis quantity may change over time and in relation to specific events. This is indicated by incremental increases in the average quantity of cannabis forms including dried herb, concentrates, number of joints, and vaping cartridges, from before legalization of cannabis in Canada (October 2018), to after the declaration of the COVID-19 pandemic (March 2020). Specifically, an increasing trend was found over time in relation to the quantity of dried herb, which was the most common form used in this study. It is also interesting to note that in some cases, the quantity of use is lower at the start of each new event relative to the end of the previous one. Although joint size and the number of joints smoked on a typical day increased after legalization, an average decrease was observed from the time frame following legalization and before EVALI was declared an epidemic. However, while joint size decreased, the number of joints per day increased from before to after the declaration of the pandemic for females. These inconsistent patterns which show an average increase in relation to specific events followed by a decline suggest that cannabis consumption behaviors may be associated with changing social and cultural contexts, events, and life trajectory (negative events) among EAs. Additional longitudinal research is needed to further explore the varying patterns of cannabis use over time.

Although THC-containing vaping products have been found to play a key role in the propagation of the EVALI epidemic [37,61], our study suggests an increasing trend in THC-based-vaping-product use among EAs. Thus, the EVALI outbreak may not have negatively influenced vaping consumption. This is in line with recent evidence from the US showing that information regarding the role of THC-based vaping oils, and vitamin E acetate, did not significantly change risk perceptions surrounding vaping products in current users [61]. Furthermore, compared to the US, EVALI cases remain substantially lower in Canada, which may explain our study findings of increased average vaping quantity. Correspondingly, cannabis vapers in Canada may have been less concerned given that most cases in the US were linked to illicit products [62], while legalization within Canada brought greater access to legal products which passed through stricter provisions for testing [63]. Indeed, recent evidence suggests that smoking has declined, while vaporizing has increased since 2020 [52].

In line with the literature, cannabis vaping may lead to an increase in quantity over time, as vaping becomes a consistent and repetitive behavior, and leads to more puffs being taken during each session, thus limiting the time that one vaping cartridge might last [57]. Consequentially, vaping high potency cannabis extracts—commonly reported among EAs [64]– may magnify the risk of developing CUD, due to a higher likelihood of building tolerance over time and experiencing withdrawal symptoms [24,65]. In fact, recent studies have shown that at least 7% to 29% of college students have used vaping devices for cannabis [39,66,67]. The popularity of vaping among young adults may be related to a number of reasons, such as the discreetness of the vapor, addition of flavors that are incorporated into the vaping liquids, the versatility of devices, and the rapid onset of desirable effects [65,66,67,68]. The increased usage of vaping products may have important clinical implications, as research remains in its infancy regarding the respiratory health effects in relation to its compared advantages and disadvantages to methods of cannabis combustion [68,69,70], although emerging evidence suggests various health risks associated with vaping cannabis [69,70].

The increase in quantity of cannabis used after legalization found in this study, may have been a result of developed interest for the effects and engagement in the liberty of cannabis laws that allowed accessibility to diverse strains and forms [23]. Following through the timeline, while subsequent public health precautions may have encouraged the cessation of smoking or vaping based on risks associated with respiratory illnesses (EVALI and COVID-19), psychosocial-associated risk factors brought forth due to the COVID-19 pandemic, such as lockdowns and isolation inducing acute and chronic stressors [41], may have positively influenced consumption, as suggested by the increase in the average quantity of cannabis smoked and vaped in this study.

During the COVID-19 pandemic, studies focused on the quantity of joints used in a day, suggesting boredom, stress, and loneliness as motives of increased substance use [51,71]. Specifically, in line with findings from a study in the Netherlands [51], we found an increase in the average number of joints after the declaration of the pandemic. In Belgium, Vanderbruggen et al. [71] did not find any changes in the number of joint days smoked per day, although individuals reported increases in smoking tobacco and alcohol. Increases in the quantity of cannabis smoked may be particular to age-specific groups, as the mean age of individuals was 42.1 (14.6) years in the Belgium study [71], whereas the other study examined changes in a younger cohort (mean years: 32.7) [51]. In line with other research, our results show that with regard to the COVID-19 pandemic, cannabis consumption patterns were influenced, with more frequent herbal cannabis users consuming more cannabis, and infrequent users consuming less on average [72]. In addition, our findings suggest that frequent use is associated with greater consumption on average, which is reflected by differences in average quantity of use reported between heavy and light users [10,73].

It is important to note that although research indicates an increase in the prevalence of females using cannabis, more research is required to delineate the effects of specific forms of cannabis used in relation to sex-specific mechanisms [14]. For example, the quantity of consumption may vary in relation to sex based on the amount of cannabis needed to achieve desired effects. The quantity of consumption is an important indicator of psychoactive effects, which is dependent upon the dose used [74,75]. Quantity measures in this study may be lower for females compared to males since physiologically, females need to smoke a lower quantity and thus need a lower dose of THC inhaled to achieve the same acute effects as males [75]. These findings highlight the importance of clinical implications in examining biological mechanisms contributing to sex differences in the acute effects of THC, and in relation to varying cannabis exposures (e.g., quantity, frequency, methods, forms).

### Strengths and Limitations

This study sheds light on cannabis-use trends in relation to single-item indicators by examining specific measures in relation to the quantity of various cannabis forms where limited research exists. These findings reveal unique aspects of the amount consumed of various cannabis forms in relation to sex and may aid in determining adverse or harm-reduction measures of consumption. Continued efforts to examine quantity can have important implications for the prevention and treatment of cannabis-related problems. In addition, to our knowledge, this is the first study to examine how quantity measures of multiple cannabis forms may change in relation to legalization, EVALI, and COVID-19 by sex-specific analyses in Canada. At the same time, this paper contributes to establishing standardized criteria for measuring cannabis, which would promote uniformity in quantifying its consumption, thereby aiding the development of research and clinical assessment standards. This, in turn, would enable clinicians to better identify individuals who are at a higher risk of acute and/or chronic issues resulting from cannabis use by gaining insights into how consumption patterns affect health outcomes. Additionally, considering gender-specific factors is important, as significant epidemiological and clinical distinctions have been observed between males and females in relation to cannabis use.

Despite these strengths, there are limitations to consider. First, asking participants to retrospectively report cannabis consumption behaviors in relation to specific measurements of quantity may have introduced recall bias, and the events of legalization, EVALI and COVID-19 were primed in the survey as trigger points for recall during data collection. Second, our study recruited individuals that were 18 years old who were minors during the time of cannabis becoming legalized in Canada (approximately 16 years old); therefore, they may have experienced different changes in cannabis use than respondents of age. In addition, the sample size was relatively small and largely from Ontario.

Due to the cross-sectional exploratory design of this study, we cannot make any causal inferences. Therefore, it is unclear if the events had a direct effect on the quantity of consumption. Similarly, any changes in relation to cannabis quantity consumed post-legalization of recreational cannabis use in Canada (17 October 2018) may be attributed to legalization and not necessarily reflect the subsequent public heath events, including the outbreak of EVALI and the COVID-19 pandemic. Differences in length between the time periods is also a limitation, given that the mean consumption in one period may not reflect exactly the same process and has the same meaning in other periods. Nonetheless, there seems to be a slow adoption of incremental increases in consumption post-legalization, which was followed by successive events that are relevant to cannabis use and may have influenced users to adapt their consumption. In addition, experimental use is prone to influence the natural progression of increased consumption, as users may transition to more frequent and daily use over time [34]. Furthermore, the awareness of EVALI was limited in this study and more likely to have affected individuals that vape. Alternatively, EVALI may have influenced users to adopt forms of cannabis that can be smoked rather than vaped, thus leading to an increase in dried herb.

Moreover, infrequent, or varying use, and sharing—which is common among cannabis users—can limit the reported capacity [3,54]. It is important to note that our study asked participants to report their quantity personally used, which may indicate that users are personally increasing their quantity of use over time. In addition, participants were able to choose more than one form to report as their primary form of use. Therefore, changes in quantity may be due to switching from one form to another, or using non-combustion or vaping methods (i.e., orally administered forms) during the course of time examined in this study. Indeed, cannabis users tend to use more than one form regularly [3,15], which makes it difficult to accurately examine patterns of quantity consumption for one specific form or method of consumption. The level of THC content used for each form in this study by participants were not examined, which may have important implications in relation to public health concerns if cannabis users are adapting their THC content along with the quantity used.

In addition, more people reported using the herb in its dried form and in the form of joints than any other form. This may have limited the power to examine significant changes in quantity for other forms in this study, including, hash and concentrates. Due to power considerations, we were not able to stratify by sex for changes in relation to hash quantity. Finally, examining vaping quantity is difficult as there are a diverse range of vaping devices, forms of cannabis used in vaping devices, as well as the size of cartridges and tanks used [57].

## 5. Conclusions

Given the substantial changes (all within a relatively short time span) in relation to cannabis policy, corresponding health risks associated with cannabis products and its derivatives, and emerging respiratory illnesses (EVALI and COVID-19) that make cannabis users that smoke or vape particularly vulnerable, examining changes in consumption can provide timely evidence of health-risk behaviors. Greater access to cannabis may increase the quantity of use over time and subsequently build tolerance, particularly for heavy users. Therefore, the quantity of consumption may be critical in addressing the progression of cannabis use disorder, as well as the relationship between measures of consumption and physiological health risks. Sex-specific differences in cannabis smoking and vaping behaviors underscore the need for gender- and sex-focused research as sub-groups may respond differently to factors that may influence cannabis use. Further research is needed to examine the predictors of changes in cannabis consumption behaviors and how these patterns may be influenced by the complex interactions of biological and psychosocial factors.

## Figures and Tables

**Figure 1 ijerph-20-06213-f001:**
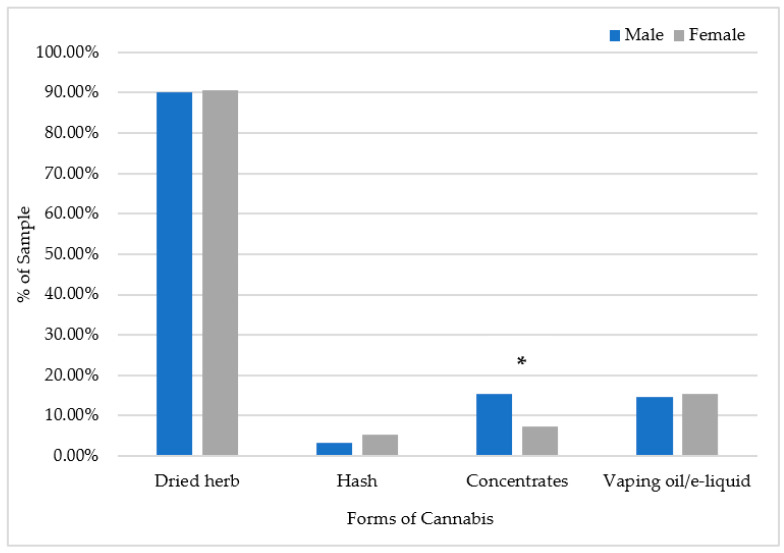
Forms of cannabis indicated as the primary form used predominantly or most often by males and females (n = 312). * *p* < 0.05.

**Table 1 ijerph-20-06213-t001:** Average quantity of cannabis use for the total sample population and stratified by sex, before and after three events: legalization of recreational cannabis use in Canada, EVALI, and the COVID-19 pandemic.

	Legalization		EVALI		COVID-19		*p*-Value
	Before October 2018	Since October 2018	Before November 2019	Since November 2019	Before March 2020	Since March 2020	
Forms (units)	M (SD)	M (SD)	M (SD)	M (SD)	M (SD)	M (SD)	
Dried Herb (grams)							
Total n = 177	1.1 (1.8)	1.3 (1.1)	1.3 (1.1)	1.3 (1.1)	1.4 (2.1)	1.5 (2.3)	<0.001 ^‡^
Males n = 76	1.6 (0.23)	1.7 (2.3)	1.7 (2.3)	1.7 (2.3)	1.7 (2.3)	1.9 (2.6)	0.41 ^†^
Females n = 101	0.67 (1.2) ^a^	1.0 (1.7) ^a^	1.0 (1.7)	1.1 (1.7)	1.1 (1.9)	1.2 (2.1)	0.004 ^†^
Joint Size (grams)							
Total n = 83	0.61 (0.43)	0.74 (0.36)	0.68 (0.39)	0.73 (0.36)	0.69 (0.38)	0.66 (0.43)	<0.001 ^‡^
Males n = 26	0.72 (0.41)	0.81 (0.34)	0.75 (0.35)	0.82 (0.31)	0.76 (0.32)	0.76 (0.39)	0.17 ^†^
Females n = 57	0.56 (0.43)	0.71 (0.37)	0.64 (0.40)	0.68 (0.37)	0.66 (0.40)	0.62 (0.44)	0.049 ^†^
Number of Joints							
Total n = 84	0.77 (0.88)	0.98 (0.97)	0.88 (0.94)	0.96 (0.97)	0.90 (0.91)	1.1 (1.3)	<0.001 ^‡^
Males n = 27	1.2 (1.1)	1.5 (1.2)	1.3 (1.2)	1.6 (1.2)	1.3 (1.1)	1.4 (1.5)	0.049 ^†^
Females n = 57	0.58 (0.70)	0.72 (0.70)	0.66 (0.70)	0.68 (0.69)	0.72 (0.78)	0.86 (1.1)	0.006 ^†^
Hash (grams)							
Total n = 13	0.42 (0.52)	0.52 (0.54)	0.40 (0.26)	0.33 (0.19)	0.25 (0.23)	0.33 (0.30)	<0.001 ^‡^
Males n = 4	0.38 (0.25)	0.69 (0.38)	0.44 (0.13)	0.44 (0.13)	0.44 (0.13)	0.44 (0.13)	0.15 ^b^
Females n = 9	0.44 (0.62)	44 (0.61)	0.39 (0.31)	0.28 (0.195)	0.17 (0.22)	0.28 (0.341)	0.39 ^b^
Concentrates (grams)							
Total n = 28	0.32 (0.45)	0.41 (0.31)	0.44 (0.43)	0.43 (0.31)	0.46 (0.42)	0.57 (0.51)	<0.001 ^‡^
Males n = 16	0.39 (0.54)	0.47 (0.35)	0.53 (0.51)	0.50 (0.33)	0.56 (0.49)	0.63( 0.51)	0.40 ^†^
Females n = 12	0.23 (0.29)	0.33 (0.25)	0.31 (0.26)	0.33 (0.27)	0.33 (0.27)	0.50 (0.53)	0.11 ^†^
Vapes (number of cartridges)							
Total n = 46	0.35 (0.80)	0.71 (1.1)	0.74 (1.2)	0.82 (1.1)	0.82 (1.3)	1.1 (1.5)	<0.001 ^‡^
Males n = 17	0.50 (1.2)	0.97 (1.3)	1.2 (1.5)	1.0 (1.3)	1.2 (1.9)	1.4 (2.0)	0.29 ^†^
Females n = 29	0.26 (0.46)	0.55 (0.99)	0.47 (0.82)	0.69 (0.97)	0.62 (0.66)	0.87 (1.1)	0.048 ^†^

Note: ^‡^ Bayesian Estimates of group means using repeated measures analysis of variance (ANOVA) indicates statistical significance within-subject differences for total sub-sample of each form used. ^†^ General linear models using repeated measures ANOVA for within-subject differences for females and males using Greenhouse–Geisser correction. Means in the rows denoted by the same subscript ^a^ are significantly different from one another (*p* < 0.05) as indicated by a Bonferroni post hoc test. ^b^ Inconclusive evidence to further stratify sex.

**Table 2 ijerph-20-06213-t002:** Average monthly quantity of cannabis forms used in relation to each time period.

	Legalization		EVALI		COVID-19		*p*-Value ^†^
	Before October 2018	Since October 2018	Before November 2019	Since November 2019	Before March 2020	Since March 2020	
Form (units)	M (SD)	M (SD)	M (SD)	M (SD)	M (SD)	M (SD)	
Dried Herb (grams)							
Total n = 173	15.75 (44.01)	22.58 (50.70)	23.39 (55.84)	26.72 (58.50)	28.62 (58.50)	34.44 (67.56)	<0.001
Males n = 73	24.15 (56.20)	30.17 (61.45)	31.92 (66.41)	35.42 (70.57)	36.19 (70.44)	39.24 (73.94)	0.097
Females n = 100	9.62 (31.39) ^ab^	17.05 (40.59)	17.18 (46.02)	20.36 (47.19)	23.09 (53.07) ^a^	30.94 (62.65) ^b^	<0.001
Joints (grams)							
Total n = 80	11.42 (29.95)	14.62 (32.97)	13.83 (33.02)	14.36 (32.88)	15.24 (31.85)	21.25 (43.61)	0.003
Males n = 26	21.98 (43.80)	27.15 (47.29)	26.23 (47.73)	27.18 (47.26)	23.63 (42.05)	36.05 (58.94)	0.083
Females n = 54	6.33 (18.73)	8.59 (21.24)	7.85 (20.95)	8.19 (20.93)	11.20 (25.03)	14.13 (32.21)	0.022
Vapes (number of cartridges)							
Total n = 44	0.83 (2.83)	1.97 (4.44)	4.44 (18.29)	4.81 (18.27)	2.64 (5.84)	10.03 (33.47)	0.179
Males n = 16	1.33 (4.30)	2.55 (5.04)	3.63 (5.78)	3.64 (5.78)	4.49 (8.50)	17.06 (51.82)	0.277
Females n = 28	0.55 (1.49)	1.63 (4.11)	4.90 (22.61)	5.48 (22.62)	1.58 (3.31)	6.00 (15.68)	0.384

Note: ^†^ General linear models using repeated measures ANOVA for within-subject differences for females and males using Greenhouse–Geisser correction. Means in the rows denoted by the same subscript ^ab^ are significantly different from one another (*p* < 0.05) as indicated by a Bonferroni post hoc test.

**Table 3 ijerph-20-06213-t003:** Quantity of cannabis used across four time periods stratified by light and heavy users.

	Before Legalization (October 2018)	Between Legalization and EVALI (October 2018 to November 2019)	Between EVALI and COVID-19 (November 2019 to March 2020)	Since COVID-19 (March 2020)	*p*-Value ^†^
Forms (unit)	M (SD)	M (SD)	M (SD)	M (SD)	
Dried Herb (grams)					
Light user (n = 37)	0.42 (1.15)	0.61 (1.42)	0.54 (0.91)	0.43 (0.77)	0.617
Heavy user (n = 138)	1.24 (1.91) ^abc^	1.50 (2.03) ^a^	1.59 (2.14) ^b^	1.79 (2.49) ^c^	<0.001
Joints (grams)					
Light user (n = 36)	0.32 (0.33)	0.34 (0.29)	0.32 (0.28)	0.28 (0.39)	0.507
Heavy user (n = 44)	0.99 (1.21) ^abc^	1.22 (1.26) ^a^	1.25 (1.20) ^b^	1.61 (1.67) ^c^	<0.001
Vapes (number of vaping cartridges)					
Light user (n = 26)	0.23 (0.48) ^ab^	0.70 (1.03)	0.76 (0.74) ^a^	1.11 (1.11) ^b^	0.005
Heavy user (n = 12)	0.81 (1.31)	1.19 (1.28)	1.39 (1.70)	1.69 (2.27)	0.360

Note: ^†^ General linear models using repeated measures ANOVA for within-subject differences for females and males using Greenhouse–Geisser correction. Means in the rows denoted by the same subscript ^abc^ are significantly different from one another (*p* < 0.05) as indicated by a Bonferroni post hoc test.

## Data Availability

The data presented in this study are available on request from the corresponding author. The data are not publicly available due to ongoing research.

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
