# Peer review of "Changes in Quantity Measures of Various Forms of Cannabis Consumption among Emerging Adults in Canada in Relation to Policy and Public Health Developments"

_ijerph, 2023, doi:10.3390/ijerph20136213_

Round 1

Reviewer 1 Report

This manuscript reports findings from a cross sectional survey on changes in cannabis use behavior among young adults (18 to 29 y) in Canada. The intro is centered on North American literature, which is fine because recent changes in legislation have occurred there. Furthermore the health damages related to the “vaping epidemic” seem to be a US phenomenon. This is supported by only 25% in your sample who were aware about EVALI. In the discussion, the authors refer to a somewhat larger extent to Dutch and Belgian literature.

The authors report self-reported increases in cannabis consumption over time, most pronounced in females and for vaping. From the study design, it remains unclear if this is a “natural course” of cannabis use among experienced users (which was an inclusion criterion) where increase occurs with higher tolerance over time. This aspect is picked up in the limitations – and there is correctly not drawn any causal interference between the three events studied: legalization, EVALI epidemic and covid-19 pandemic.

I suggest three improvements for you to consider:

- there is no mentioning of age besides 18 to 29 and I wonder if you did not ask for age at study inclusion, or merely did not report it? The former should be mentioned as limitation, the latter needs to be corrected.

- In table 1, please include the specific times for the three events in the headline, month and year of onset should be enough.

- You start out the limitations section with a couple of sentences on “strengths”, please change the headline to “strengths and limitations” to reflect the content.

Reviewer 2 Report

Please find the review in the file attached below

Reviewer 3 Report

I would like to thank the authors for the job they did on the manuscript. The article, although looks like having publication potential, has numerous issues that must be addressed. I will outline them for each part of the article. But in the meanwhile, I suggest rejection.

My main concern in this study was the ability to properly capture the real effects of the changes between the studied periods and to ensure that the findings properly reflect the entire periods to which the study was divided. As known, when there is a major health-related, policy, or social change which can directly or/and indirectly impact substance use, the initial period is typically characterized by moderate-up-to-sharp increase in consumption (because of the expected shortage in supply, restrictions, or lockdowns, and more). In the end of this period, there is some tendency toward normalization of the substance use behavior. Given that the time frames between most of the studied periods are quite short, it is difficult to claim that the mean of consumption in one period reflects exactly as same process and has as same meaning as in other periods. What is more, the periods themselves are unequal in terms of their duration. For instance, the "Since EVALI" period lasts about four months, whereas the "Prior to legislation" period (which, in itself, may be subdivided in any possible way) could last for decades. In addition, the "Since EVALI" period comes after two very significant periods and can be influenced by them (therefore, not being totally exogenous). The authors are required to address this issue, possibly by reconsidering the current division. 

Abstract

Lines 14-16 - The sentence appearing in these lines should be rewritten as it is difficult to understand what is meant here.

Keywords: Please arrange them in alphabetical order. In addition, many of them seem unrelated to the current study (like "public health"). Please reconsider them (in any case, there is no need for such an abundant keyword list).

Introduction

General comments:

1. Although this part provides important information, it totally lacks a comparison of its content with the research in the field. What are the issues/elements/characteristics that differ from the previous studies on COVID-19 and cannabis? Or particularly gender and cannabis? In which aspects this study is better than the previous ones? Please elaborate better here.

2. The importance of the study is not provided. Why should we understand gender differences in the studied phenomena? Please elaborate better also here.

3. The literature is very much Canada-based. Since the article's audience is expected to be international, more related research from outside Canada should be discussed. For example: on gender differences: Burdzovic Andreas et al. (2020), Cabanillas-Rojas (2020), Cuttler et al. (2016), Carliner et al. (2017); on COVID-19 and cannabis: Boehnke et al. (2021), Lake et al. (2022), Brenneke et al. (2022), Bonny-Noach et al. (2022). Please incorporate these articles.

Specific comments:

Line 69 - I think you missed/forgot the citation of Jones et al. (2016) in your count. Please address this.

Lines 69-70 - Why may vaping pose notable health risks to people? Please explain this in the text.

Lines 100-102 - Indeed, a study by Sznitman et al. (2022) found that secondary stressors were associated, both directly and indirectly, with increased cannabis use during COVID-19. Please cite this work in the current regard.

Lines 103-105 - Indeed. Cannabis sharing patterns too underwent some changes, including during the course of the pandemic. The study by Rosenberg and Sznitman (2022) clearly shows this. Please cite this work. 

Methods

Line 143 - If the respondents could choose more than one "primary" form, how can you define it as "primary"? Which one will be "more primary" than others? Please elaborate here.

Lines 170-178 - Please outline the dependent variables in a more accurate way so that they would be clearly distinguished one from the other.

Lines 183-184 - Why only these two variables were controlled for? Please provide the explanation in the text.

Line 189 - Why differences with respect to frequencies were also analyzed using Chi-square?

Results

Line 233 - Where the H1 was outlined? Please ensure that you do not insert sentences that have no previous appearance.

Line 240 - Please replace the word "small" with "borderline".

Line 242 - Please rewrite the "moderate evidence for no difference" part of the sentence.

Line 250 - Again, please replace "small" with "borderline".

Limitations

You have much more limitations than you have outlined. First, the sample is Toronto area-based. Second, the sample is small. Third, only people with two administration modes were studied. Fourth, only people with good English command participated, whereas substance use is typically more prevalent in ethnic/immigrant minority populations. Please incorporate them in the text.

A very good quality
